# Highly Permeable Matrimid^®^/PIM-EA(H_2_)-TB Blend Membrane for Gas Separation

**DOI:** 10.3390/polym11010046

**Published:** 2018-12-30

**Authors:** Elisa Esposito, Irene Mazzei, Marcello Monteleone, Alessio Fuoco, Mariolino Carta, Neil B. McKeown, Richard Malpass-Evans, Johannes C. Jansen

**Affiliations:** 1Institute on Membrane Technology, ITM-CNR, Via P. Bucci 17/C, 87036 Rende (CS), Italy; e.esposito@itm.cnr.it (E.E.); m.monteleone@itm.cnr.it (M.M.); a.fuoco@itm.cnr.it (A.F.); 2Department of Chemistry, Durham University, Stockton Road, Durham DH1 3LE, UK; irenemazzei.94@gmail.com; 3Department of Chemistry, College of Science, Swansea University, Grove Building, Singleton Park, Swansea SA2 8PP, UK; 4EastChem, School of Chemistry, University of Edinburgh, David Brewster Road, Edinburgh EH9 3FJ, UK; Neil.McKeown@ed.ac.uk (N.B.M.); R.Malpass-Evans@ed.ac.uk (R.M.-E.)

**Keywords:** polymer blend, polymer of intrinsic microporosity, PIM-EA(H_2_)-TB, Tröger’s base polymer, Matrimid, gas separation, mixed gas diffusion

## Abstract

The effect on the gas transport properties of Matrimid^®^5218 of blending with the polymer of intrinsic microporosity PIM-EA(H_2_)-TB was studied by pure and mixed gas permeation measurements. Membranes of the two neat polymers and their 50/50 wt % blend were prepared by solution casting from a dilute solution in dichloromethane. The pure gas permeability and diffusion coefficients of H_2_, He, O_2_, N_2_, CO_2_ and CH_4_ were determined by the time lag method in a traditional fixed volume gas permeation setup. Mixed gas permeability measurements with a 35/65 vol % CO_2_/CH_4_ mixture and a 15/85 vol % CO_2_/N_2_ mixture were performed on a novel variable volume setup with on-line mass spectrometric analysis of the permeate composition, with the unique feature that it is also able to determine the mixed gas diffusion coefficients. It was found that the permeability of Matrimid increased approximately 20-fold with the addition of 50 wt % PIM-EA(H_2_)-TB. Mixed gas permeation measurements showed a slightly stronger pressure dependence for selectivity of separation of the CO_2_/CH_4_ mixture as compared to the CO_2_/N_2_ mixture, particularly for both the blended membrane and the pure PIM. The mixed gas selectivity was slightly higher than for pure gases, and although N_2_ and CH_4_ diffusion coefficients strongly increase in the presence of CO_2_, their solubility is dramatically reduced as a result of competitive sorption. A full analysis is provided of the difference between the pure and mixed gas transport parameters of PIM-EA(H_2_)-TB, Matrimid^®^5218 and their 50:50 wt % blend, including unique mixed gas diffusion coefficients.

## 1. Introduction

Significant progress has been made in the development of new polymers for the fabrication of gas separation membranes. Polymers used for gas separation membranes can be either rubbers, generally characterized by a relatively low selectivity but high permeability and thus enabling a great productivity, or glassy polymers, characterized by low permeability but high gas-pair selectivity and offering a good separation efficiency. This trade-off between permeability and selectivity, first introduced [1], and updated by Robeson [2] and then by Pinnau et al. [3] is typical for polymeric membranes, and was theoretically explained by Freeman et al. [4]. An exceptional novel class of so-called polymers of intrinsic microporosity (PIMs), introduced by Budd, McKeown and co-workers [5], which combine an exceptionally high permeability with relatively high selectivities, is responsible for the large upward shift of the upper bound in recent years. PIMs owe their exceptional behaviour to extremely rigid [6] and highly contorted [7,8] polymer chains, which do not allow efficient packing and is responsible for a high free volume in these materials.

While most high performing PIMs require complex and expensive synthetic procedures, the key challenge to improve the competitiveness of membrane separations over other gas separation techniques is the fabrication of inexpensive polymer membranes, having a good trade-off between permeability and selectivity. Commercial polymers used for gas separation membranes generally have a high selectivity, but low permeability. Matrimid^®^5218, a commercial amorphous glassy polyimide, is one of those polymers, and improvements of its overall performance would necessarily require an increase in its permeability. One option to improve the permeability of a polymer is by blending with another polymer to combine synergistically the best properties of the two individual materials [9,10]. This is not an easy task, because due to only a very small gain in entropy, most polymers are not miscible at the molecular level. Early studies on the miscibility of Matrimid^®^5218 were reported by Grobelny et al. [11], and since then different polymers were blended with Matrimid, in order to increase its separation performance [12,13,14]. The blend of Matrimid^®^5218 with polyethersulfone (PES) yielded mechanically stable flat film [15] and hollow fiber [16] membranes for efficient CO_2_/N_2_ separation; its blend with polysulfone (PSf) enhanced the stability of Matrimid in CO_2_/CH_4_ binary mixtures, due to the mitigation of CO_2_ plasticization [17]. While the CO_2_ and CH_4_ permeability of polysulfone/Matrimid^®^5218 blend membranes increase with the polyimide content, there is an optimum in the CO_2_/CH_4_ mixed gas selectivity (≃30) for the blend with 20% of Matrimid [18].

With their unique properties, starting with PIM-1 [19] that defined the 2008 Robeson upper bound [2], numerous new PIMs with a wide range of different chemical structures [6,20,21,22,23,24,25,26,27,28] and increasingly efficient gas separation performance, have moved the Robeson upper bounds further for several gas pairs [3]. Despite their high permeability, the sophisticated synthesis and high costs hinder PIMs from being the basis for large scale membrane production and industrial applications at present [29]. In addition, for demanding separations, their modest selectivity needs improvement. The idea underpinning the present work is to blend PIMs with a highly selective commercial polymer such as Matrimid^®^5218, in order to tailor their permeability and selectivity. In the last few years, the archetypal PIM-1 was blended with several other commercial polymers, especially with highly selective polymers such as polysulfone [30] and polyimide [31] in order to increase its selectivity. Blends of PIM-1 with polyphenylenesulfone (PPSU) and sulfonated polyphenylenesulfone (sPPSU) exhibited similar permeability of the polysulfone but enhanced selectivity compared to the neat PIM-1, with an additional anti-plasticization effect under mixture conditions [30]. Similarly, the blending of carboxylated PIM-1 (cPIM-1) with the highly selective co-polyimide P84 demonstrated increased selectivity with a simultaneous reduction of the permeability, as the amount of P84 was increased in the blend [31]. The first blend of PIM-1 with Matrimid was made by Yong et al. and they also studied the miscibility of Matrimid and Torlon with cPIM-1 [32,33,34]. Addition of a small quantity of Matrimid in PIM-1 improved the O_2_/N_2_ separation performance, while a small amount of PIM-1 in an excess of Matrimid enhanced the CO_2_/CH_4_ gas separation performance. Moreover, they used the PIM-1/Matrimid blend to fabricate hollow fibers, demonstrating the greater versatility of the blend for obtaining an ultrathin dense layer, potentially suitable for industrial use [35]. More recently, a novel blend membrane of PIM-1 and a Tröger’s Base (TB) polymer showed lower pure gas permeability but higher ideal selectivity than that of the pristine PIM-1 membrane [36].

In the present paper, we study the properties of a blend membrane based on the highly permeable PIM-EA(H_2_)-TB [37,38,39,40] and the highly selective Matrimid^®^5218 (Figure 1). PIM-EA(H_2_)-TB is a member of the new class of PIMs consisting of Tröger’s Base and ethanoanthracene (EA), forming a particularly rigid polymer backbone. The equivalent polymer with methyl substituents on the two bridgehead positions of the ethanoanthracene unit, PIM-EA-TB, was shown to have a marked size-sieving behaviour that favours the diffusion of gases with smaller kinetic diameters, and surpasses the 2008 Robeson upper bound for the O_2_/N_2_, H_2_/CH_4_ and H_2_/CO_2_ gas pairs [8,41]. The PIM-EA(H_2_)-TB is highly selective, as the analogous PIM-EA-TB, and was recently used to increase the hydrogen permeability of polybenzimidazole by blending [40].

The aim of this work to enhance the permeability of Matrimid by the addition of PIM-EA(H_2_)-TB, and to find the desired combination of the high permeability of the PIM and the high selectivity of the polyimide. Detailed analysis of the gas transport parameters under single and mixed gas permeation conditions provides deep insight into the role of gas diffusivity and solubility in the overall transport properties of the novel Matrimid^®^5218/PIM-EA(H_2_)-TB blend. In particular, a novel mixed gas permeation setup with the unique possibility to determine the mixed gas diffusion coefficients will provide unprecedented information on the coupling effect between CO_2_ and CH_4_ or CO_2_ and N_2_ during permeation of the respective mixtures in the neat polymers and the blend.

## 2. Materials and Methods

### 2.1. Materials

Matrimid^®^5218 was kindly supplied by Huntsman (Basel, Switzerland) and PIM-EA(H_2_)-TB was prepared as described previously [37] and the purified polymer, isolated as a powder, was used without any further treatment.

### 2.2. Preparation of Matrimid^®^5218/PIM-EA(H_2_)-TB Blend Membranes

The casting solutions of both pure polymers were prepared at a concentration of 2 wt % of the polymer in dichloromethane (DCM). Homogenous solutions were obtained under magnetic stirring overnight. The blend solution was prepared by mixing equal amounts of the two individual solutions. The blend membrane containing 50 wt % of both Matrimid^®^5218 and PIM-EA(H_2_)-TB was prepared pouring the appropriate amount of the blend solution into a metallic casting ring of 3 cm diameter, placed on a Teflon^®^ support. The solvent was evaporated at room temperature for 24 h, yielding optically defect-free membranes Figure 1.

### 2.3. Membranes Characterization

Chemical and morphological analysis of membranes were performed by scanning electron microscopy (SEM) on a Phenom Pro X desktop SEM, equipped with backscattering detector (Phenom-World B.V., Eindhoven, The Netherlands) and infrared spectroscopy (FTIR) analyses on a Spectrum Spotlight Chemical Imaging Instrument (PerkinElmer). Single gas permeation tests were carried out at 25 °C and at a feed pressure of 1 bar, using a fixed-volume pressure increase instrument (ESSR, Geestchacht, Germany), described elsewhere [43]. Permeability coefficients, *P*, and diffusion coefficients, *D*, were determined by the time-lag method [44]. The simplest model of permeation through dense polymeric films describes permeability as the product of diffusion coefficient and solubility coefficient. Thus, the apparent solubility, *S*, was indirectly calculated as *S* = *P*/*D*. The ideal selectivity is the ratio of permeability of two species, α_(A/B)_ = *P*_A_/*P*_B_. Mixed gas permeation tests were carried out using a custom made constant pressure/variable volume instrument, described elsewhere [45,46], equipped with a quadrupole mass filter (HPR-20 QIC Benchtop residual gas analysis system, Hiden Analytical). Measurements were carried out as a sequence of increasing and subsequently decreasing pressure steps in the range from 1–6 bar(a).

## 3. Results and Discussion

### 3.1. Chemical and Morphological Characterization

The chemical interaction between Matrimid^®^5218 and PIM-EA(H_2_)-TB was studied by means of FTIR-ATR. The IR-spectrum for neat Matrimid^®^5218, neat PIM-EA(H_2_)-TB and for the blend were shown in Figure 2. The distinctive imide peaks of Matrimid^®^5218 appear at 1712 for C=O stretching and at 1361 cm^−1^ for C–N stretching, these peaks are also found in the spectrum of the blend. The characteristic peaks of PIM-EA(H_2_)-TB are the CH_2_ asymmetric stretch vibrations of the ethanoanthracene (EA) unit at 2960 cm^−1^ and the scissoring vibrations at 1420 cm^−1^. The water peak (3370 cm^−1^) in the neat PIM-EA(H_2_)-TB and in the blend demonstrate the relatively hydroscopic nature of TB-PIMs [23]. The good compatibility of Matrimid^®^5218 and PIM-EA(H_2_)-TB was also deduced from the high optical transparency of the film and high mechanical resistance. The cross-sectional SEM image shows very few separate domains (less than 1% of the area) confirming the excellent miscibility between PIM-EA(H_2_)-TB and Matrimid (Figure 2).

### 3.2. Pure Gas Transport Properties

Single gas permeation measurements were carried out in the order He, H_2_, O_2_, N_2_, CH_4_ and CO_2_ at 25 °C on time-stabilized membranes [47]. Figure 3 shows a plot of the gas transport parameters of 30 days aged PIM and blend membranes, and a >1 year aged Matrimid membrane as a function of the membrane composition. The quantitative values are reported in Table 1. The experimental results highlight the different behaviour of Matrimid^®^5218 and PIM-EA(H_2_)-TB. For all gases, the pure gas permeability (Figure 3a), diffusivity (Figure 3c) and solubility (Figure 3e) are all higher in the Matrimid^®^5218/PIM-EA(H_2_)-TB blend than in neat Matrimid membrane, thanks to the higher free volume of the PIM.

The gas permeability and diffusivity are greater for the neat PIM, while there is little difference in solubility between the blend and the neat PIM. The effect of composition is largest for diffusivity, which increases about two orders of magnitude from Matrimid to PIM, whereas the solubility increases less than an order of magnitude. The time lag of H_2_ and He is too short to be measured accurately in the relatively thin neat Matrimid^®^5218 membrane, and thus the related diffusion coefficient and solubility of this membrane could not be determined. The combined effect of S and D is reflected in an exceptionally high permeability of the neat PIM membrane, with an approximately three orders of magnitude higher N_2_ and CH_4_ permeability compared to the Matrimid membrane. On the other hand, the selectivity is generally higher in Matrimid^®^5218 (Figure 3b and Table 1), especially for gas pairs with very different kinetic diameters, like H_2_/N_2_, He/N_2_, mainly as a result of the much higher diffusion selectivity (Figure 3d). The O_2_/N_2_ selectivity is higher in Matrimid^®^5218 due to a slightly higher diffusion selectivity and solubility selectivity, whereas the higher CO_2_/N_2_ selectivity in Matrimid^®^5218 must be ascribed mainly to the higher solubility selectivity (Figure 3e).

The blend membrane with 50 wt% of each polymer exhibits intermediate properties with respect to the two individual polymers, with a roughly linear trend in permeability on a logarithmic scale. This trend suggests that the two polymers have good compatibility and form a homogeneous blend, because Robeson [48] anticipated that that the permeability, P_b_, of a homogeneous polymer blend can be expressed as:(1)$$\ln P_{b}=\;\phi_{1}\ln P_{1}+\;\phi_{2}\ln P_{2}$$ where *ϕ*_1_ and *ϕ*_2_ are the volume fractions of the two polymers in the blend, and P_1_ and P_2_ are their respective permeabilities. The minor deviations of the experimental data from linearity are likely due to the use of the weight fraction (Figure 3) instead of volume fraction (Equation (1)). The volume fraction of the PIM, which has a lower density than Matrimid^®^5218, is higher than the weight fraction. To some extent, the nonlinearity may also be due to slight differences in the degree of physical ageing in the three samples, typically observed for PIMs but less in common glassy polymers with lower free volume.

### 3.3. Mixed Gas Transport Properties

#### 3.3.1. Mixed Gas Permeability

The membrane performance for two relevant industrial separations was investigated via mixed gas permeability measurements on the pristine Matrimid^®^5218, PIM-EA(H_2_)-TB and on the Matrimid^®^5218/PIM-EA(H_2_)-TB blend (See Appendix A
Table A1). Measurements were performed from 1 to 6 bar(a) with two binary gas mixtures of CO_2_/CH_4_ (35:65 vol %) and CO_2_/N_2_ (15:85 vol %), in order to simulate the biomethane purification process from biogas and CO_2_ capture from flue gas, respectively (Figure 4, Table 2).

The Matrimid^®^5218 membrane showed very weak pressure-dependence of permeability and selectivity for both gas pairs, CO_2_/CH_4_ and CO_2_/N_2_ in the given pressure range. It does not show significant hysteresis between the pressure increase steps and the pressure decrease steps, which means that neither substantial physical ageing, nor CO_2_ induced dilation of the polymer takes place. On the other hand, for the Matrimid^®^5218/PIM-EA(H_2_)-TB blend and the neat PIM-EA(H_2_)-TB membranes, the CO_2_ permeability decreases as function of the feed pressure for both gas mixtures. This is typical for polymers with distinct dual mode behaviour, which is indeed very common for PIMs [49]. The N_2_ permeability decreases in a similar fashion as a function of pressure, and thus the CO_2_/N_2_ selectivity is virtually constant in all three samples. In the blend and the pure PIM, the methane permeability is slightly less pressure-dependent than that of CO_2_, and therefore the CO_2_/CH_4_ selectivity decreases in these two membranes. The stronger effect of CO_2_ on methane than on nitrogen is likely due to its higher concentration (35 vol %) in the CO_2_/CH_4_ mixture, compared to 15 vol % in the CO_2_/N_2_ mixture. All membranes were aged for at least a month before the measurements, so that further ageing during the experiments should not be expected. Only neat PIM-EA(H_2_)-TB showed a slightly lower permeability in the pressure decrease steps, but the difference was hardly more than the experimental error, so may not be necessarily attributed to physical ageing. On the contrary, the Matrimid^®^5218/PIM-EA(H_2_)-TB blend showed weak hysteresis for CO_2_ and CH_4_, with slightly higher permeability of both gases in the pressure decrease steps. This suggests a slight dilatation of the polymer matrix and/or removal of trace amounts of residual solvent at elevated CO_2_ partial pressures.

#### 3.3.2. Mixed Gas Diffusion and Solubility

As described recently, a unique feature of our mixed gas permeation setup with on-line mass-spectrometric analysis of the permeate composition is that it allows the determination of, not only, the permeability but also the diffusion coefficients of the individual gases in the mixture [45,46]. The results for the three membranes are given in Table 3. The diffusion coefficient is determined by a time lag method for gas mixtures [45], and the solubility is approximated indirectly as *S* = *P/D*. The most obvious result for the blend and PIM membrane is that, while the CH_4_ and N_2_ permeability change relatively little in the mixture (Table 2), their diffusion coefficients increase substantially and since the change in diffusion coefficient of CO_2_ is much smaller, the diffusion selectivity decreases significantly. The gas solubility shows exactly the opposite trend: the CH_4_ and N_2_ solubility decrease in the presence of CO_2_. This provides clear evidence of competitive sorption by CO_2_ at the expense of the less condensable gases.

### 3.4. Robeson Plots and Comparison with Literature Blend-Data

The gas permeability data of neat Matrimid^®^5218, PIM-EA(H_2_)-TB, and the blended membrane are plotted in the Robeson diagrams for CO_2_/N_2_, CO_2_/CH_4_, O_2_/N_2_ and H_2_/CH_4_ (Figure 5). From Matrimid^®^5218 to the blend and to pure PIM-EA(H_2_)-TB, the diagrams show a strong increase in the pure gas permeability, accompanied by a modest decrease in ideal selectivity, as a common trend for all gas pairs. Literature data of pure Matrimid and other PIM/Matrimid blend membranes for CO_2_/N_2_ and CO_2_/CH_4_ separation are plotted for reference. Most of the data of our samples lie inside the data cloud near its top range. The cloud of Matrimid data derives from the different measurement conditions, membrane preparation and conditioning.

Only the CO_2_/N_2_ selectivity of the neat PIM and the blend lie significantly higher than the cloud and these samples are positioned much closer to the most recent upper bound than Matrimid. Especially for CO_2_/N_2_, with PCO_2_ = 198 Barrer and αCO_2_/N_2_ = 29, the Matrimid^®^5218/PIM-EA(H_2_)-TB blend presents a better trade-off between permeability and selectivity compared to other Matrimid^®^5218/PIM (50:50) blends reported in the literature, such as PIM-1/Matrimid (PCO_2_ = 155 Barrer; αCO_2_/N_2_ = 27) and cPIM-1/Matrimid (PCO_2_ = 145 Barrer; αCO_2_/N_2_ = 24) [32,34]. Of note is that the Matrimid permeability was increased approximately 20-fold by the addition of 50 wt % of PIM-EA(H_2_)-TB. This offers the potential for the preparation of asymmetric or thin film composite membranes with much higher permeability, without compromising selectivity too much.

## 4. Conclusions

The pure and mixed gas permeation measurements of neat polymer of intrinsic microporosity PIM-EA(H_2_)-TB, Matrimid^®^5218 and their 50/50 wt % blend demonstrated that the permeability of Matrimid^®^5218 can be increased dramatically by the addition of the PIM whilst maintaining a reasonably high selectivity. Mixed gas permeation experiments reveal a comparable CO_2_ permeability and a slightly higher selectivity than the ideal one for the CO_2_/CH_4_ and CO_2_/N_2_ gas pairs, and addition of the PIM moves the performance of Matrimid^®^5218 closer to the Robeson upper bound for these gas pairs. Interestingly, in spite of the slightly higher mixed gas permselectivity, the mixed gas diffusion selectivity is significantly lower than the ideal diffusion selectivity. This indicates that the presence of CO_2_ favours the diffusion of the slower gases N_2_ and CH_4_, but that the effect of competitive sorption, which reduces their solubility, dominates the overall performance of the membranes. All membranes show a moderate to weak decrease in either the permeability or the selectivity for the CO_2_/CH_4_ and CO_2_/N_2_ gas pairs with increasing pressure, typical for dual mode sorption behaviour. The very strong effect of PIM-EA(H_2_)-TB offers the possibility to tailor the permeability of Matrimid^®^5218 over a wide range and opens perspectives for making high permeability thin film composite membranes with the mechanical resistance of Matrimid and PIM-like permeabilities.

## Figures and Tables

**Figure 1 polymers-11-00046-f001:**
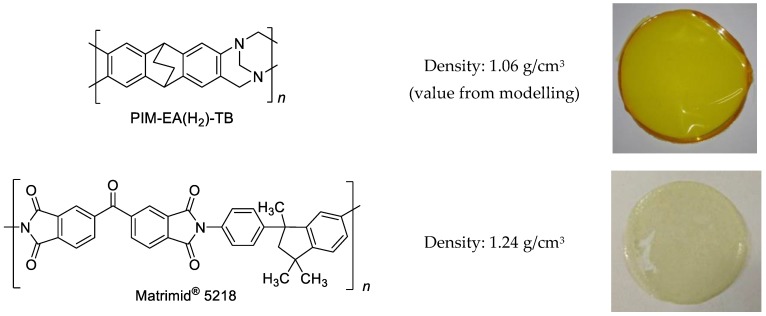
Structure, properties and membrane sample of PIM-EA(H_2_)-TB and Matrimid. Density from Ref. [37] and [42], respectively.

**Figure 2 polymers-11-00046-f002:**
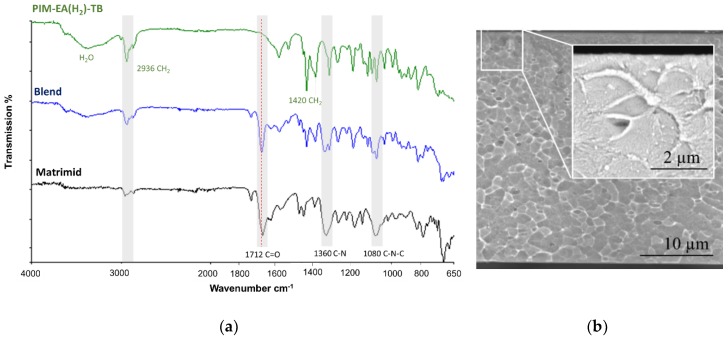
(**a**) ATR-FTIR spectra of the Matrimid^®^5218/PIM-EA(H_2_)-TB blend and neat polymer membranes, and (**b**) SEM cross-sectional image of the Matrimid^®^5218/PIM-EA(H_2_)-TB blend membrane at a magnification of 8000× and 30,000× at an accelerating voltage of 10 kV.

**Figure 3 polymers-11-00046-f003:**
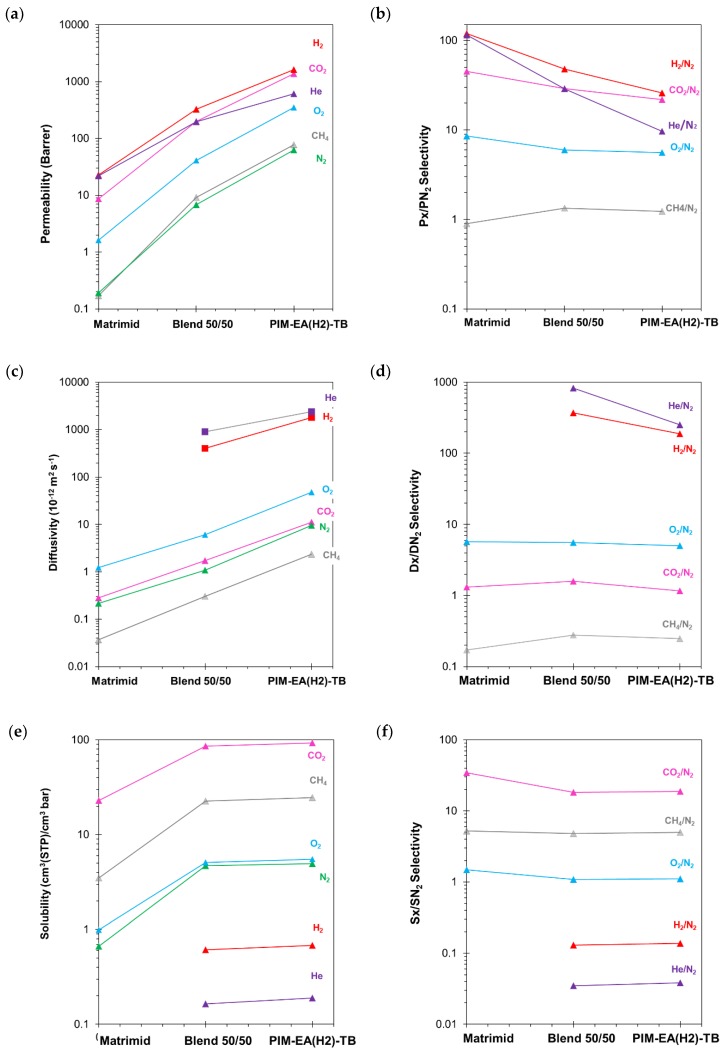
(**a**) Permeability, (**c**) diffusivity and (**e**) solubility coefficients with their respective selectivities (**b**,**d**,**f**) of Matrimid^®^ (100%); Matrimid^®^5218/PIM-EA(H_2_)-TB blend and PIM-EA(H_2_)-TB (100%). The lines are indicated as a guide to the eye.

**Figure 4 polymers-11-00046-f004:**
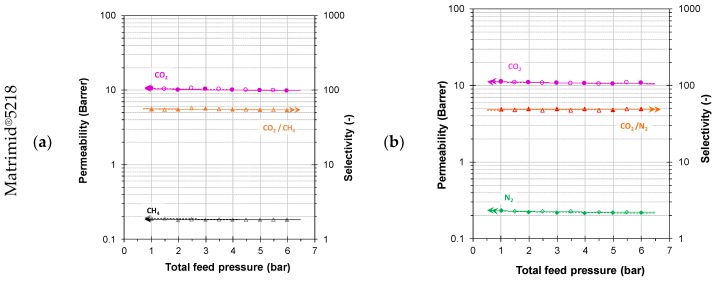
(**a**,**c**,**e**) Pressure dependence of CO_2_ and CH_4_ permeabilities and CO_2_/CH_4_ selectivity using the binary mixture of CO_2_/CH_4_ (35:65 vol %) for neat polymers and Matrimid^®^5218/PIM-EA(H_2_)-TB blend membranes. (**b**,**d**,**f**) Pressure dependence of CO_2_ and N_2_ permeabilities and CO_2_/N_2_ selectivity in binary mixtures CO_2_/N_2_ (15:85 vol %); Closed symbols are used for the stepwise increase of the pressure and open symbols for the subsequent stepwise decrease of the pressure.

**Figure 5 polymers-11-00046-f005:**
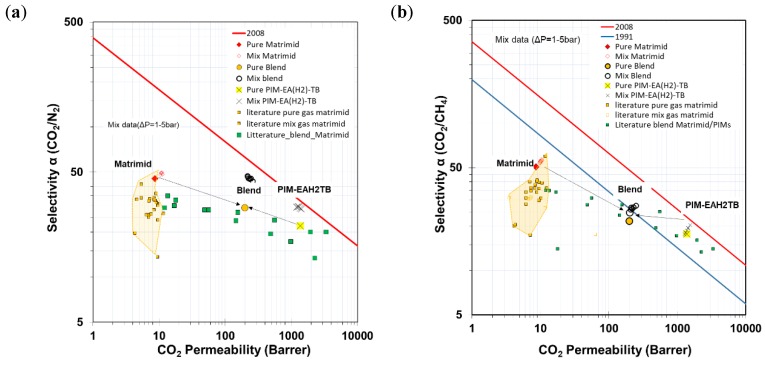
Robeson’s plots of ♦ neat 
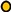
Matrimid^®^5218, X PIM-EA(H_2_)-TB and Matrimid^®^5218/PIM blend membranes for CO_2_/N_2_ (**a**), CO_2_/CH_4_ (**b**), O_2_/N_2_ (**c**) and H_2_/CH_4_ (**d**) gas pairs (pure gas filled symbols and Mixed gas open symbols). Reference data for other matrimid-based blends (■) and other PIM-based blends (■) are plotted for comparison.

**Table 1 polymers-11-00046-t001:** Experimental data of single gas permeation in the neat polymers and the Matrimid^®^5218/PIM-EA(H_2_)-TB blend membrane.

	**Permeability (Barrer)**						**Selectivity (Px/Py)**						
**Membrane**	N_2_	CH_4_	O_2_	He	H_2_	CO_2_	H_2_/N_2_	He/N_2_	O_2_/N_2_	CH_4_/N_2_	O_2_/CH_4_	CO_2_/CH_4_	CO_2_/N_2_
Matrimid	0.19	0.17	1.63	21.9	22.8	8.62	120	115	8.54	0.90	9.54	50.6	45.3
Blend	6.83	9.14	41.0	197	328	198	48.0	28.8	6.00	1.34	4.48	21.6	29.0
PIM-EA(H_2_)-TB	62.8	77.6	350	606	1630	1380	25.9	9.64	5.57	1.23	4.51	17.7	21.9
	**Diffusivity (10^−12^ m^2^/s)**						**Diffusion selectivity (Dx/Dy)**						
**Membrane**	N_2_	CH_4_	O_2_	He	H_2_	CO_2_	H_2_/N_2_	He/N_2_	O_2_/N_2_	CH_4_/N_2_	O_2_/CH_4_	CO_2_/CH_4_	CO_2_/N_2_
Matrimid	0.21	0.04	1.23			0.28			5.74	0.17	33.6	7.69	1.31
Blend	1.09	0.30	6.04	900	403	1.73	370	826	5.54	0.28	19.9	5.71	1.59
PIM-EA(H_2_)-TB	9.6	2.36	47.9	2400	1790	11.2	187	251	5.01	0.25	20.3	4.72	1.17
	**Solubility (cm^3^(STP)/cm^3^ bar)**						**Solubility selectivity (Sx/Sy)**						
**Membrane**	N_2_	CH_4_	O_2_	He	H_2_	CO_2_	H_2_/N_2_	He/N_2_	O_2_/N_2_	CH_4_/N_2_	O_2_/CH_4_	CO_2_/CH_4_	CO_2_/N_2_
Matrimid	0.67	3.49	0.99			22.9			1.49	5.24	0.28	6.58	34.4
Blend	4.70	22.6	5.08	0.16	0.61	85.7	0.13	0.03	1.08	4.81	0.22	3.79	18.2
PIM-EA(H_2_)-TB	4.93	24.6	5.47	0.19	0.68	92.5	0.14	0.04	1.11	5.00	0.22	3.76	18.8

1 Barrer = 10^−10^ cm^3^ (STP) cm cm^−2^ s^−1^ cmHg^−1^.

**Table 2 polymers-11-00046-t002:** Comparison of the pure and mixed gas permeability of the neat Matrimid^®^5218, PIM-EA(H_2_)-TB and their blend membrane as measured in the mixed gas setup. Feed gas: binary mixture CO_2_/CH_4_ (35/65 vol %) and binary mixture CO_2_/N_2_ (15/85 vol %) at 1 bar.

		Permeability (Barrer)			Px/Py Selectivity (−)	
		CO_2_	CH_4_	N_2_	CO_2_/CH_4_	CO_2_/N_2_
Matrimid^®^5218	Pure gas	10.4	0.20	0.24	52	43.3
	Mix (CO_2_/CH_4_)	10.5	0.19	-	55	-
	Mix (CO_2_/N_2_)	11.3	-	0.23	-	49.1
Blend	Pure gas	198	9.1	6.83	21.66	28.99
	Mix (CO_2_/CH_4_)	250	9.09	-	27.49	-
	Mix (CO_2_/N_2_)	260	-	6.00	-	43.37
PIM-EA(H_2_)-TB	Pure gas	1391	62.6	53.1	22.22	26.20
	Mix (CO_2_/CH_4_)	1527	76.1	-	20.07	-
	Mix (CO_2_/N_2_)	1445	-	49.4	-	29.25

**Table 3 polymers-11-00046-t003:** Comparison of the pure and mixed gas diffusivity and solubility of the neat Matrimid^®^5218, PIM-EA(H_2_)-TB, and their blend membrane as measured in the mixed gas setup. Feed gas: binary mixture CO_2_/CH_4_ (35/65 vol %) and binary mixture CO_2_/N_2_ (15/85 vol %) at 1 bar.

		**Diffusivity (10^−12^ m^2^ s^−1^)**			**Dx/Dy Selectivity (−)**	
		**CO_2_**	**CH_4_**	**N_2_**	**CO_2_/CH_4_**	**CO_2_/N_2_**
Matrimid^®^5218	Pure gas	0.35	0.05	0.28	7.04	1.24
	Mix (CO_2_/CH_4_)	0.29	0.06	-	5.12	-
	Mix (CO_2_/N_2_)	0.24	-	0.30	-	0.80
Blend	Pure gas	2.36	0.46	1.58	5.71	1.59
	Mix (CO_2_/CH_4_)	2.10	0.66	-	3.18	-
	Mix (CO_2_/N_2_)	1.40	-	2.11	-	0.61
PIM-EA(H_2_)-TB	Pure gas	10.7	2.31	9.22	4.63	1.16
	Mix (CO_2_/CH_4_)	10.6	3.61	-	2.94	-
	Mix (CO_2_/N_2_)	7.14	-	10.5	-	0.68
		**Solubility (cm^3^(STP) cm^−3^ bar^−1^)**			**Sx/Sy Selectivity (−)**	
		**CO_2_**	**CH_4_**	**N_2_**	**CO_2_/CH_4_**	**CO_2_/N_2_**
Matrimid^®^5218	Pure gas	22.2	3.00	0.64	7.39	34.8
	Mix (CO_2_/CH_4_)	27.0	2.50	-	10.8	-
	Mix (CO_2_/N_2_)	34.9	-	0.57	-	61.5
Blend	Pure gas	85.8	22.6	4.70	3.79	18.3
	Mix (CO_2_/CH_4_)	89.2	10.3	-	8.66	-
	Mix (CO_2_/N_2_)	138	-	1.94	-	71.0
PIM-EA(H_2_)-TB	Pure gas	97.5	20.3	4.32	4.80	22.3
	Mix (CO_2_/CH_4_)	109	15.7	-	6.91	-
	Mix (CO_2_/N_2_)	148	-	3.53	-	42.0

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
