# Peer review of "Highly Permeable Matrimid®/PIM-EA(H2)-TB Blend Membrane for Gas Separation"

_polymers, 2018, doi:10.3390/polym11010046_

Round 1

Reviewer 1 Report

This is an interesting work backed by sound experimentation. However, the manuscript needs a revision in presentation of data before publishing. Following these comments and suggestion will help the authors to improve the manuscript quality for publication:

Authors need to provide magnified SEM images for the top layer of the dense film cross section. It seems the dense film has porous structure. Is there any explanation for this structure?

Are there any SEM images for the neat polymer derived dense film for comparison?

It seems the selectivity has some benefit (increased 7) after blended with PIM-EA(H2)-TB while the permeance decreased from 1500 to 250 Barrer, with comparison of neat PIM-EA(H2)-TB. How to access the trade-off for the blending?

Author Response

Response to Reviewer 1 Comments

This is an interesting work backed by sound experimentation. However, the manuscript needs a revision in presentation of data before publishing. Following these comments and suggestion will help the authors to improve the manuscript quality for publication:

Authors need to provide magnified SEM images for the top layer of the dense film cross section. It seems the dense film has porous structure. Is there any explanation for this structure?

an image is now added as an insert of the cross section

Are there any SEM images for the neat polymer derived dense film for comparison?

Dense films of neat polymer do not show any interesting features in SM (of course!) and therefore we did not include them in the manuscript. The reason to include the image of the blend is because we would have seen phase separation if the polymers were not miscible.

It seems the selectivity has some benefit (increased 7) after blended with PIM-EA(H2)-TB while the permeance decreased from 1500 to 250 Barrer, with comparison of neat PIM-EA(H2)-TB. How to access the trade-off for the blending?

The scope of the work is the opposite, improve the permeability of Matrimid, maintaining acceptable selectivity and good mechanical properties.

Reviewer 2 Report

Herein, the authors present the synthesis and gas separation performance of polymer membranes. The authors have examined 3x membranes, a Matrimid®5218 membrane, a PIM-EA(H2)-TB membrane and a 50/50 blend of both. The 50/50 blend was found to increase the permeability 20-fold compared to the Matrimid®5218 membrane. The work presented is of a high standard with significant experimental work that supports their conclusions, and therefore I would recommend for publication in Polymers pending the authors addressing the following:

1. While the authors and previous studies claim that PIMs contain intrinsic microporosity, it might be of interest to the reader to provide some additional evidence to support this? i.e. 195K CO2 and/or 77K N2 gas sorption analysis on a portion of each membrane.

2. The authors should ensure typo's etc. are kept to a minimum. i.e. line 28 in abstract, N2 and CH4 should be subscript.

Author Response

Response to Reviewer 2 Comments

Herein, the authors present the synthesis and gas separation performance of polymer membranes. The authors have examined 3x membranes, a Matrimid®5218 membrane, a PIM-EA(H2)-TB membrane and a 50/50 blend of both. The 50/50 blend was found to increase the permeability 20-fold compared to the Matrimid®5218 membrane. The work presented is of a high standard with significant experimental work that supports their conclusions, and therefore I would recommend for publication in Polymers pending the authors addressing the following:

1. While the authors and previous studies claim that PIMs contain intrinsic microporosity, it might be of interest to the reader to provide some additional evidence to support this? i.e. 195K CO2 and/or 77K N2 gas sorption analysis on a portion of each membrane.

Sorption is generally not measured on films because of problems with the kinetics of the process. Indeed, this would require extremely long equilibration times for every pressure step. For the same reasons, no sorption data are reported for any of the referenced manuscripts on PIM-based blends. The preparation of a powder requires precipitation of the diluted polymer solution in a nonsolvent, which would alter the microporosity. Therefore what the reviewer asks is technically not possible.

In any case, we have no reason to believe that the microporosity would not have an intermediate value between that of the two neat polymers.

2. The authors should ensure typo's etc. are kept to a minimum. i.e. line 28 in abstract, N2 and CH4 should be subscript.

We have checked the manuscript again and made a few corrections.

Reviewer 3 Report

The manuscript is well organized with many significant data. I recommend the manuscript for publication. I can give just two comments:

1) For how many membranes were the tests carried out?  What is the confidence level fulfilled by the reported results? P<0.005?

2)  The SEM image quality (fig1b) is unacceptable. Please provide another image with information about the magnification, accelerating voltage and applied sample coating material.

Author Response

Response to Reviewer 3 Comments

The manuscript is well organized with many significant data. I recommend the manuscript for publication. I can give just two comments:

1) For how many membranes were the tests carried out?  What is the confidence level fulfilled by the reported results? P<0.005?

Only one membrane sample was tested for the pure gas permeability tests and only one for the mixed gas permeation. From very thorough previous studies, we know that the precision of individual measurements is very high (maximum few % error, see for instance Ref 45 of Fraga et al, and the SI of  Ref 7 of Rose et al.). This is orders of magnitude lower than the differences between the three samples discussed here. Therefore the conclusions would not change if we introduced an error margin).

2)  The SEM image quality (fig1b) is unacceptable. Please provide another image with information about the magnification, accelerating voltage and applied sample coating material.

The missing information about the SEM image is added to the image and to the caption. The quality is dictated by the use of the backscattering detector, which was used because it would emphasize differences in composition, which a secondary emission detector would not give. The samples were not coated, so that the backscattering detector would not have any interference from the coating layer and would only depend on the sample morphology and composition.

Round 2

Reviewer 1 Report

Authors have addressed all of my comments.